



# A case study of Kuroshio Extension Front: evolution,
# structure, diapycnal mixing and instability
Jiahao Wang[1,†], Xi Chen[1,†], Kefeng Mao[1,†], and Kelan Zhu[2]
[1]College of Meteorology and Oceanography, National University of Defense Technology, Nanjing,
211101, People's Republic of China
[2]NO. 61741 Army of PLA, Beijing, 100094, People's Republic of China
[†]These authors contributed equally to this work. They are the co-first authors.
*Correspondence to: Xi Chen (lgdxchxtemp@163.com)*
**Abstract.** Satellite measurements during April to June in 2019 and direct observations from 28th to
30th May in 2019 about the Kuroshio Extension Front are conducted. The former shows the front
experience a process of stable-unstable-stable state caused by the movement of the Kuroshio
Extension's second meander and a pinched-off eddy. The latter indicates the steep upward slopes of the
isopycnals tilt northward in the strong frontal zone as well as several over 100 m thick blobs of cold
and fresh water in the salinity minimum zone of North Pacific Intermediate Water. Using isopycnal
anomaly method and diapycnal spiciness curvature method, characteristic interleaving layers are shown
primarily in $\sigma_\theta$=26.3-26.9 kg/m³, which corresponds to large variations of potential spiciness in
intermediate layers. Further analysis indicates the development of thermohaline intrusions may be
driven by the double diffusive instability and the velocity anomalies. Besides, we find the turbulence
mixing attributed to symmetric instability and shear instability is very strong in intermediate layer.
**Keywords** Kuroshio Extension Front; Evolution; Structure; Diapycnal Mixing; Instability
**1 Introduction**
The Kuroshio Extension (KE) is a variable eastward inertial jet separating from the coast of Japan near
35°N in the North Pacific Ocean [*Delman et al.*, 2015; *Kawai*, 1972; *Qiu and Chen*, 2005]. Without the
constraint of coastal boundaries, it is rich in large-amplitude meanders and energetic pinched-off eddies
[*Delman et al.*, 2015; *Ji et al.*, 2018; *Qiu and Chen*, 2005] which are often associated with the sharp
subsurface front named Kuroshio Extension Front (KEF) [*Kida et al.*, 2015; *Nagai et al.*, 2015; *Nagai*
*et al.*, 2012].
The oceanic front is the boundary of different water masses and characterized by across-front contrasts
in ocean factors, such as temperature, salinity and density [*Nagai et al.*, 2015; *Wang et al.*, 2016; *Zhu et*
*al.*, 2019]. The KEF is formed by a steep upward slope of the main pycnocline tilting northward [*Kida*
*et al.*, 2015; *Nonaka et al.*, 2006]. It is strong in winter while weak in summer, and has important
impacts on the regional ecosystem, fishery and atmosphere [*Kida et al.*, 2015; *Nagai and Clayton*, 2017;
*Pauly and Christensen*, 1995].What's more, the KEF presents different state alternately on decal time
scales: a stable state with two quasi-stationary meanders and an unstable state with a convoluted path
[*Kida et al.*, 2015; *Qiu and Chen*, 2005; *Seo et al.*, 2014]. The latter state is linked with the anticyclone
eddies detached northward from the KEF [*Itoh and Yasuda*, 2010; *Kida et al.*, 2015].
In the frontal zone, strong along-isopycnal stirring [*Macvean and Woods*, 1980; *Smith and Ferrari*,


2009] and diapycnal mixing exist [*D'Asaro et al.*, 2011; *Nagai et al.*, 2012]. Among them, the double
diffusive mixing often causes lateral fluxes of heat, salt and momentum, and results in the fine-scale
structures indicated by changes in the sign of vertical temperature or salinity gradients, known as the
thermohaline intrusions [*Ruddick and Kerr*, 2003; *Itoh et al.*, 2016; *Jan et al.*, 2019; *Nagai et al.*, 2015;
*Nagai et al.*, 2012; *Richards and Banks*, 2002; *Ruddick and Richards*, 2003; *Shcherbina et al.*, 2009;
*Stern*, 1967], while the turbulent mixing and horizontal stirring impede the intrusions [*Ruddick and
Richards*, 2003]. These processes affect the maintenance and variation of the oceanic front as well
[*Jing et al.*, 2016; *Wang and Li*, 2012]. Besides, water mass formation and subduction linked with
cabbeling and double diffusion may occur in the frontal zone [*Rudnick and Luyten*, 1996; *Talley and
Yun*, 2001].
The structure and variability of KEF has been investigated widely through recognizing sea surface
temperature and sea surface height by remote sensing measurements [*Nakano et al.*, 2018; *Jing et al.*,
2019; *Nagai and Clayton*, 2017; *Yu et al.*, 2016; *Wang and Liu*, 2015; *Wang et al.*, 2016], as well as
model outputs [*Jing et al.*, 2019; *Nagai and Clayton*, 2017; *Nonaka et al.*, 2006; *Taguchi et al.*, 2009].
However, field observations could offer higher spatial resolution and more reliable data to investigate
the KEF, but they are still rare to date. The fine-scale structures of temperature, salinity, density and
velocity, and related marine processes of KEF have not been well understood.
In this work, we investigate the evolution, structure, diapycnal mixing characteristics and instability of
the KEF based on the field observation at the end of May in 2019 and the satellite measurements during
April to June of 2019. This paper is organized as follows. Section 2 describes the data and methods
used; section 3 discusses evolution of surface thermal KEF, thermohaline and velocity structure across
the KEF, mechanisms for the thermohaline intrusions, double diffusion mixing and turbulence mixing
across the KEF, and instability of the KEF; section 4 offers conclusions.
**2 Data and Methods**
**2.1 Satellite Remote Sensing Data**
The daily satellite data sets with 1/4°× 1/4° resolution including sea surface temperature (SST),
absolute dynamic topography (ADT), sea level anomaly (SLA) and sea surface geostrophic velocities
during the end of April to the end of June in 2019 are used in this study. SST comes from Optimum
Interpolation Sea Surface Temperature (OISST) product distributed by National Oceanic and
Atmospheric    Administration    (NOAA)    (http://www.ncei.noaa.gov/data/sea-surface-temperature-
optimum-interpolation/access/avhrr-only/), and the others are from Archiving, Validation, and
Interpretation of Satellite Oceanographic (AVISO) product (http://marine.copernicus.eu/services-
portfolio/access-to-products/).
**2.2 In Situ Observations**
A hydrographic survey with four observation sections for the frontal zone is carried out from 28th to
30th May, 2019 (Figure 1k). The details of the stations could be found in Table 1. The temperature,
conductivity, and pressure are measured using a Moving Vessel Profile (MVP) 300-3400 instrument
(1m-vertical intervals). We smooth the row profiles with a 5-point (5m) running mean. The velocity
profiles along the ship track are obtained by an OS-300 kHz Acoustic Doppler Current Profiler (ADCP)
(2m-bin size) and a MARINE 38 kHz ADCP (16m-bin size). In order to obtain high quality flow field
data, we merge the data of two ADCPs: using 300kHz ADCP data for the current shallow than 75m,



and using 38kHz data for the current below than 75m. Finally, we obtain the data set including
temperature, salinity and current shallow than 500m.

| Section | Location | Heading direction | Number of stations |
|---------|----------|-------------------|--------------------|
| A1 | 151.74°-151.53°E, 38.11°-39.19°N | Southeast to Northwest | 21 |
| A2 | 151.06°-151.32°E, 39.17°-38.14°N | Northwest to Southeast | 21 |
| A3 | 151.17-150.61°E, 38.12°-39.46°N | Southeast to Northwest | 27 |
| A4 | 149.73-150.50°E, 39.26°-38.13°N | Northwest to Southeast | 28 |

Table 1. Details of Sections A1-A4, the number of stations mean the number of MVP stations set for
each section.
**2.3 Methods**
In this study, a gradient-based algorithm is utilized for the SST fields [*Yuan and Talley*, 1996]. The
surface thermal front could be identified by the horizontal SST gradient in each geo-referenced grid.
The SST gradient magnitude ($GM_T$) is defined by the following formula:

$$GM_T = |\nabla_H T| = \sqrt{(\frac{\partial T}{\partial x})^2 + (\frac{\partial T}{\partial y})^2}$$

We calculate several parameters based on the in situ observations as follows:
In the practically orthogonal potential density-potential spicity (σ-π) coordinate system, water mass and
isopycnal layer analysis can be carried out accurately. We calculate potential spicity by the least square
method. The detailed procedure is basically the same as that described in *Huang et al.* [2018]. After
that, when we make thermohaline analysis, we convert potential temperature-salinity (θ-S) coordinate
system to σ-π coordinate system, as shown in Figure 3.
We characterize thermohaline intrusions through two methods. One is isopycnal anomaly method:
using isopycnal salinity (interpolate salinity into 0.01 kg/m³-interval isopycnal) anomaly S˙ as an
indicator of the intrusion strength, where the anomaly is computed relative to some ''mean background
state'' of the ocean (in this paper, it is calculated through 13-point (0.13kg/m³) running mean)
[*McDougall*, 1987; *Shcherbina et al.*, 2009]. The other is diapycnal spiciness curvature method: using
the second derivative of potential spiciness with respect to potential density $\tau_{\sigma\sigma}$ as an indicator of water
mass interleaving [*Shcherbina et al.*, 2009].
In order to examine double diffusive instability, the Turner angle Tu is calculated from the profiles of
potential temperature θ and salinity S as

$$T_u = \tan^{-1}(\frac{\alpha\theta_z + \beta S_z}{\alpha\theta_z - \beta S_z})$$

where α and β are thermal expansion and haline contraction coefficients, respectively [*Ruddick*, 1983].


We also assess the diapycnal mixing including double diffusion mixing and turbulence mixing as
follows:
for the former, *Nagai et al.* [2015] observe double diffusive convection below the main stream of the
KE, compare their results with the previous parameterizations for double diffusion, and recommend
parameterization from *Radko et al.* [2014] for salt fingering regime while parameterization from
*Fedorov* [1988] for diffusive convection regime. In this paper, we also use these parameterizations of
effective thermal diffusivity ($K_\theta$):
In the salt fingering regime with the density ratio $R_\rho > 1$ ($R_\rho = \dfrac{\alpha \theta_z}{\beta S_z}$):

$$K_\theta = F_s K_t \gamma$$

where $F_s = a_s(R_\rho-1)^{-0.5}+b_s$, $\gamma = a_g exp(-b_g R_\rho)+c_g$, $a_s=135.7$, $b_s=-62.75$, $a_g=2.709$, $b_g=2.513$, $c_g=0.5128$;
In the diffusive convection regime with the density ratio $0<R_\rho<1$:

$$K_\theta = 0.909 v exp(4.6 exp[-0.54(R_\rho^{-1} - 1)])$$

where $v$ is molecular viscosity of seawater, which takes the value $1.5 \times 10^{-7} m^2/s$.
for the latter, we use the parameterization of turbulent eddy diffusivity ($K_\rho$):

$$K_\rho = \Gamma \varepsilon N^2$$

where $\Gamma$ is the mixing efficiency, which takes the value 0.2, N is buoyancy frequency, and $\varepsilon$ is the
dissipation rate of turbulent energy calculated by Thorpe scale $L_T$. The specific calculation of $L_T$ could
be found in *Thorpe* [2005] and *Zhu et al.* [2019].
What's more, when we examine instability of frontal zone, we calculate Ertel Potential Vorticity (q),
horizontal buoyancy gradient ($\nabla_h b$) and Richardson number ($R_i$). q can be decomposed into the vertical
component $q_v$ and horizontal baroclinic component $q_h$.

$$q = q_v + q_h = (f + \zeta)N^2 + \omega_h \nabla_h b$$

where f is Coriolis parameter, $\zeta$ is the vertical relative vorticity, $\omega_h$ is the horizontal component of the
absolute vorticity $\omega$ ($\omega = f\hat{k} + \nabla \times u$), and $\nabla_h b$ could be calculated through thermal wind relation:
$\nabla_h b = f \dfrac{\partial u_g}{\partial z} \times \hat{k} = -f\omega_h.$
Therefore, $q_h$ can be expressed as

$$q_h = -\frac{|\nabla_h b|^2}{f}$$

And $R_i$ is calculated as

$$R_i = -\frac{N^2}{|\frac{\partial u_h}{\partial z}|}$$

[*Jing et al.*, 2016].
**3 Results**
**3.1 Evolution of Surface Thermal Kuroshio Extension Front from Satellite Measurements**

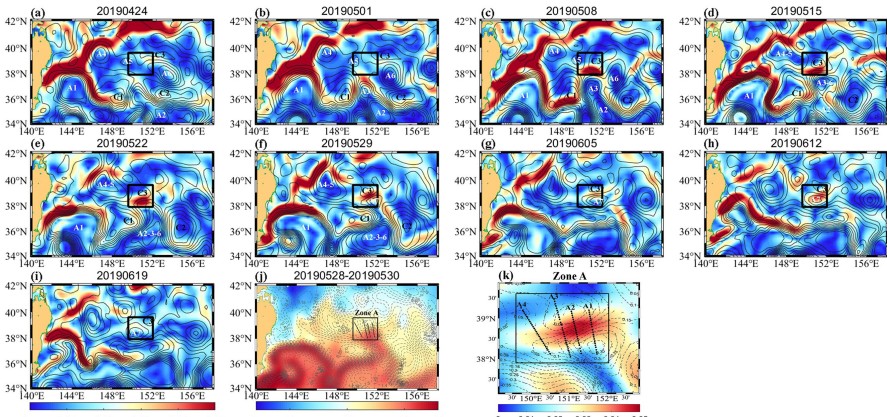

Figure 1. (a-i) Daily SST gradients (shading in ℃/km) and SLA (contours in m) east of Japan are
shown every seven days from the end of April to the end of June in 2019. Intervals for contour lines are
0.1 m. Black boxes indicate the observation area. Some eddies are labeled as follows: anticyclone
eddies (A) in white and cyclone eddies (C) in black; if two eddies merge, we named "Ax-xx" or
"Cx-xx". (j) Mean SST (shading in ℃) and ADT (contours in m) east of Japan during the observation
period. Intervals for contour lines are 0.05 m. Black box is the observation area named Zone A. Black
dots are the observation stations. (k) Mean SST gradients (shading in ℃/km), SLA (contours in m) and
geostrophic currents (vectors in m/s) in the Zone A during the observation period. Intervals for contour
lines are 0.05 m. Black dots are the observation stations. The observation sections are labeled from A1
to A4 in black.
The frontal activities east of Japan present significantly variations during the end of April to the end of
June in 2019, both temporally and spatially (Figure 1). The KEF band (>0.025°C/km) has the
characteristics of meanders in the upstream KE. Generally, it is always strong (about 0.05°C/km) from
east coast of Japan to 146°E corresponding to the first meander of KE jet, polytropic at the second
meander and always weak (about 0.025-0.03°C/km) east of the second mender. Undoubtedly, the KE
jet affects the distribution of KEF to a large extent.
Due to the variability of the second meander, the KEF varies strongly there. Satellite measurements
indicate both of them experience a process of stable-unstable-stable state. The second meander
gradually moves towards north during the end of April to the end of May. It transports the warm and
saline water masses, and mixes them with the cold and brackish water masses in Kuroshio-Oyashio
Confluence Region (KOCR). This process causes the convoluted KEF's northward movement and
enhancement (from 0.025°C/km to >0.035°C/km) as well as generates the pinched-off eddies (e.g.
A2-3-6) and merged eddies (e.g. A7) at the region from 148°E to 154°E. During the end of May to
early June, the second meander reverts to south and becomes flat; the KEF returns to stable gradually.
The crest of the second meander moves from 37°N in 24th April to the northest at 38.5°N in 22th May,
which generates the strongest part of KEF (about 0.05°C/km) located at the black box of Figure 1.
Undoubtedly, the water masses get colder in the further north (Figure 1j); therefore, the temperature
gradient between the norther KORC and KE water masses get higher. After that, an anticyclone eddy
named A7 detaches from the crest. It locks and carries the KE water mass whose SST is >20°C (Figure
1j) to maintain the intensity at the black box in 29th May. Thereafter, the anticyclone eddy A7 moves



westward and the north cyclone eddy C3 moves eastward. The SST gradient between them becomes
lower and reduces to approximately 0.025°C/km in 19th June.

**3.2 Thermohaline and Velocity Structure Across the Kuroshio Extension Front**

The shipboard observation of Zone A is made during 28th to 30th May. Satellite measurements
indicate A1-A3 sections could capture the front, the anticyclone eddy A7 and the cyclone eddy C3; A4
section could capture a small anticyclone eddy near 39°N else (Figure 1k). The tight-station settings
and high-resolution instruments could depict their thermohaline and velocity structure clearly.
The potential temperature and salinity across the front observed by the MVP show clear contrasts
between the warm and saline, and the cold and fresh waters (Figure 2). In general, A1-A3 sections'
observation shows the steep upward slopes of the isotherms, isohalines and isopycnals tilt southward
south of 38.5-38.6°N, northward from 38.5-38.6°N to 39°N and southward north of 39°N; A4 section's
observation shows the slopes tilt southward south of 38.22°N, northward from 38.22°N to 38.7°N,
southward from 38.7°N to 38.85°N, northward from 38.85°N to 38.9°N, and southward north of 39°N.
Furthermore, characteristics of the slopes reflect the eddies' and front's traits: the isolines' throughs
represent the locations nearest the warm eddy A7's center of the four sections, which are gradual to
south from A1 to A4 section, indicate A7's distribution is southwest-northeast upper than 350 m,
similarly, the crests represent the locations nearest the cold eddy C3's center of A1-A3 sections, and, in
A4 section, the isolines are relatively flat from 38.22°N to 38.8°N and rise from 38.8°N to 39°N, which
signify the A4 section capture the small warm eddy mentioned before near 39°N; the isolines' rise is
O(10) m in the south interior and is O(100) m in the north interior and exterior of eddy A7, which
suggests the difference of thermohaline properties between A7 and C3 is conspicuous while in the
eddies' the other side interior is relatively small; range of the significantly rising and sinking isolines
corresponding to the sharp horizontal gradient in potential temperature and salinity represent the frontal
zone, therefore, the front's range is 38.6-39°N in A1 and A2 section, 38.3-38.8°N in A3 section and
38.15-38.7°N in A4 section, which is consistent well with the satellite measurements.
The currents measured by the ADCPs could reflect the eddies' and front's locations as well. The cores
of the positive zonal velocities occur in the upper layers in 38.6-38.9°N of A1 section, in 38.6-39°N of
A2 section, in 38.4-38.8°N of A3 section, and in 38.4-38.7°N of A4 section, which represent the
boundaries of the eddies A7/C3 and correspond to the ranges of prominently rising isolines. The strong
frontal zone locates at the eddies' boundaries. The core of the positive zonal velocities couldn't but the
zero velocities could extend to intermediate layers, which reflects the eddy center's depth are deeper
than the boundary. Besides, although the meridional velocities are weaker than the zonal velocities in
general, they still can't be left out as the cross-frontal velocities approximately and its sloping layers
appeared to cross isopycnal surfaces which could affect the variabilities of the isopycnals.
Another prominent feature is the blobs of low salinity between $\sigma_\theta$=26.5-26.7 kg/m$^3$ of over ~100 m
thickness in north of 38.8°N in A1 section, in 38.2-38.5°N and north of 38.7°N in A2 section, in north
of 38.5°N in A3 section, and in 38.4-38.75°N in A4 section (Figure 2), which is the salinity minimum
zone of North Pacific Intermediate Water (NPIW) ($\sigma_\theta$=26.3-26.9 kg/m$^3$) [*Talley and Yun*, 2001]. The
zonal velocities suggest that NPIW is in the weak flow region and the meridional velocities suggest
that the salinity minimum zone of NPIW is extended/obstructed by cross-frontal velocities. Large
variations in potential spiciness across the KEF seen in θ-S plot and σ-π plot (Figure 3) illustrate that
interleaving layers may arise when along-isopycnal transports occur in intermediate layers [*Nagai et al.*,





2015; *Smith and Ferrari*, 2009]. We choose the single representative profile which is in the frontal
zone and also contain the salinity minimum zone from every section, as shown in gray curves in Figure
3; these gray θ-S and σ-π curves are zigzag deeper than $\sigma_\theta$=26.5 kg/m³, which are necessary anatomies
of interleaving layers, and can be seen in many other profiles.
In order to detect the thermohaline intrusions across the KEF better, we use both isopycnal salinity
anomaly method and diapycnal potential spiciness curvature method in an isopycnal coordinate system
which could reduce the distortion of interleaving features by internal waves, as shown in Figure 4.
These two methods detect the nearly unanimous interleaving layers. It is easily seen that the locations
of relatively high absolute values of S˙ and $\tau_{\sigma\sigma}$ which have spatial continuity along the isopycnal are
primarily in NPIW layers ($\sigma_\theta$=26.3-26.9 kg/m³), especially the layers contain salinity minimum zone in
the northern frontal zone, and appear stronger vertical coherence there (more full oscillations from
minimum negative to maximum positive S˙ and $\tau_{\sigma\sigma}$). The intrusions have cross-frontal orientation, are
laterally coherent for up to O(10) km, and their vertical thickness is approximately O(100) m.

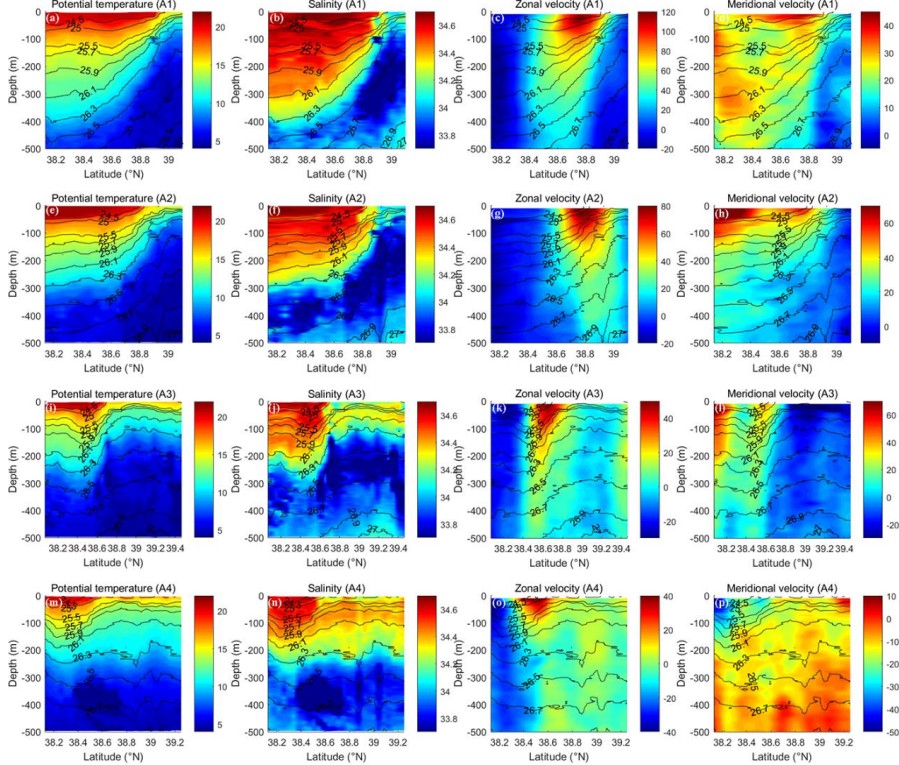

Figure 2. (a,c,i,m) Potential temperature (shading in ℃), (b,f,j,n) salinity (shading in psu), (c,g,k,o)
zonal velocity (shading in cm/s) and (d,h,l,p) meridional velocity (shading in cm/s) of the four sections.
Contours indicate the potential density (kg/m³).

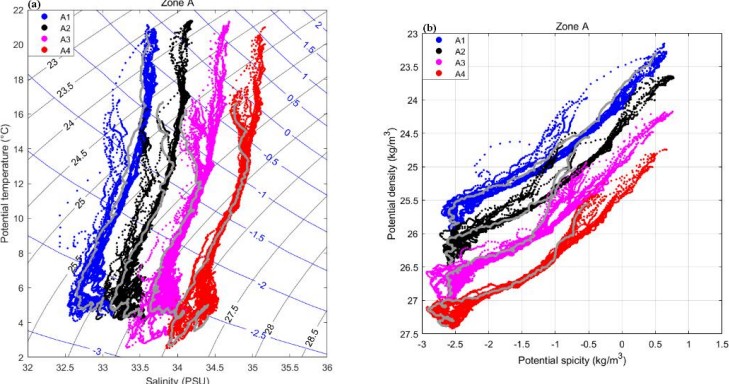

Figure 3. (a) Potential temperature–salinity (θ-S) diagram of the four sections. A1/A2/A3/A4 section's
result is shifted along the x axis: Δx=-1/-0.5/0/0.5. The gray curves indicate the representative profiles
of A1-A4 sections obtained at 38.84°N, 38.83°N, 38.76°N and 38.60°N, respectively, to show the
thermohaline intrusions. Potential density (black contours in kg/m$^3$) and potential spicity (blue contours
in kg/m$^3$) in θ-S space are also shown. (b) Potential density-potential spicity (σ-π) diagram of the four
sections. A1/A2/A3/A4 section's result is shifted along the y axis: Δσ=-1/-0.5/0/0.5. The gray curves
are the same representative profiles of (a).

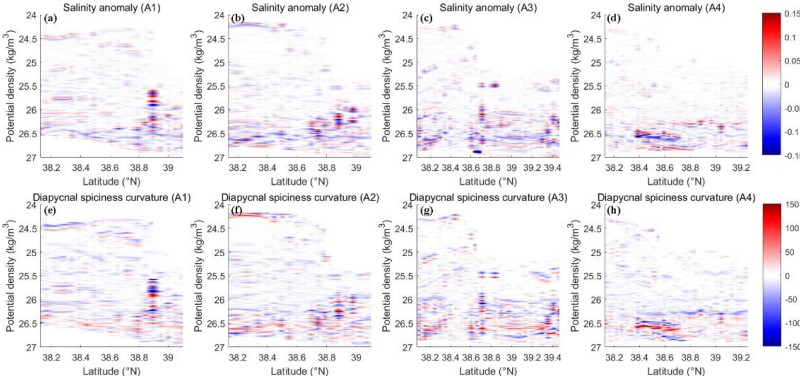

Figure 4. (a-d) Salinity anomaly (shading in psu) and (e-h) diapycnal spiciness curvature (shading in
m$^3$/kg) of the four sections.

### 3.3 Mechanisms for the Thermohaline Intrusions

We discuss the thermohaline and velocity structure across the front last section. We find the strong
front exists in the boundaries of the warm and cold eddy, and the thermohaline intrusions mostly
occurred in NPIW layers, especially the layers contain the salinity minimum zone of NPIW in the
northern frontal zone. In this section, we investigate the mechanisms for the thermohaline intrusions.

Double diffusive processes are attributed by previous studies as the driving mechanism for the growth
of intrusions through changing potential density [*McDougall*, 1985; *Talley and Yun*, 2001; *Toole and
Georgi*, 1981]. Turner angle (Tu) computed for MVP data is shown in Figure 5a-d. When 45° (72°) <



Tu < 90°, thermohaline stratification is favorable for (strong) salt fingers, when -90° < Tu < -45° (-72°)
for (strong) diffusive convection. The stratification is stable as Tu is between −45° and 45° and
gravitationally unstable as Tu is beyond ± 90° [*Ruddick*, 1983]. The value of Tu indicates that the
(strong) salt fingering regime mainly appear ($\sigma_\theta$=26.1-26.5 kg/m$^3$) upper than $\sigma_\theta$=26.5 kg/m$^3$ and the
diffusive convection regime mainly appear deeper than $\sigma_\theta$=26.7 kg/m$^3$. In $\sigma_\theta$=26.5-26.7 kg/m$^3$, the salt
fingering regime and diffusive convection regime alternately appear. Therefore, salt fingering
interfaces occur at the top and diffusive interfaces at the bottom of the intruded fresh, cold NPIW
layers; the interleaving layers prefer to the alternate salt fingering and diffusive convection interfaces.
Note that the double diffusive instability is a necessary but not sufficient condition for the generation of
interleaving layers: the growth of interleaving layers is conceivably affected by the background shear
and density gradient [*Beal*, 2007; *Jan et al.*, 2019]. In the zonal velocity core of the frontal zone, the
strong current upper than $\sigma_\theta$=26.3 kg/m$^3$ (Figure 2) and the weak salinity variation in $\sigma_\theta$=26-26.3 kg/m$^3$
(Figure 3) restrict the interleaving layers' development in a fixed section.
We also calculate the salinity anomaly, density anomaly and velocity anomaly of the four
representative profiles, as shown in Figure 6. Note that the velocity anomaly is the meridional velocity
anomaly which can be seen as the cross-frontal velocity anomaly approximately, since the intrusions
have cross-frontal orientation (Figure 4). The correlation coefficient we calculated between salinity
anomaly and density anomaly is 0.28/0.41/0.50/0.43, between salinity anomaly and velocity anomaly is
0.13/0.25/0.004/0.24 for A1/A2/A3/A4. We focus on the salinity minimum zone of NPIW: for the
profile from A1/A1/A3/A4 section, it is about 250-400/200-350/150-375/300-425 m and $\sigma_\theta$=26.5-26.7
kg/m$^3$. The correlation coefficient of the interleaving layer between salinity anomaly and density
anomaly is 0.21/0.47/0.50/0.48, between salinity anomaly and velocity anomaly is -0.35/0.65/0.68/0.29
for A1/A2/A3/A4. This imply the thermohaline intrusions may link with not only double diffusive
process of salt fingering but also velocity anomalies.
The vertical shears of the zonal (along-frontal) and meridional (cross-frontal) velocity have the same
magnitude (Figure 7). The vertical shear of along-frontal horizontal current indicates that negative
shear is very strong in the frontal zone as the boundaries of the two eddies and positive shear is very
strong in the eddies' the other side interior, which reflects the dynamic property of eddies that the
velocities increase/decrease with depth around the eddy center/boundary as well. The vertical shear of
cross-frontal horizontal current presents intense and spatially coherent fine-scale shear layer, which is
influenced mostly from high vertical wavenumber shear presumably caused by internal waves, and may
drive intrusions [*Beal*, 2007; *Itoh et al.*, 2016; *Rainville and Pinkel*, 2004].



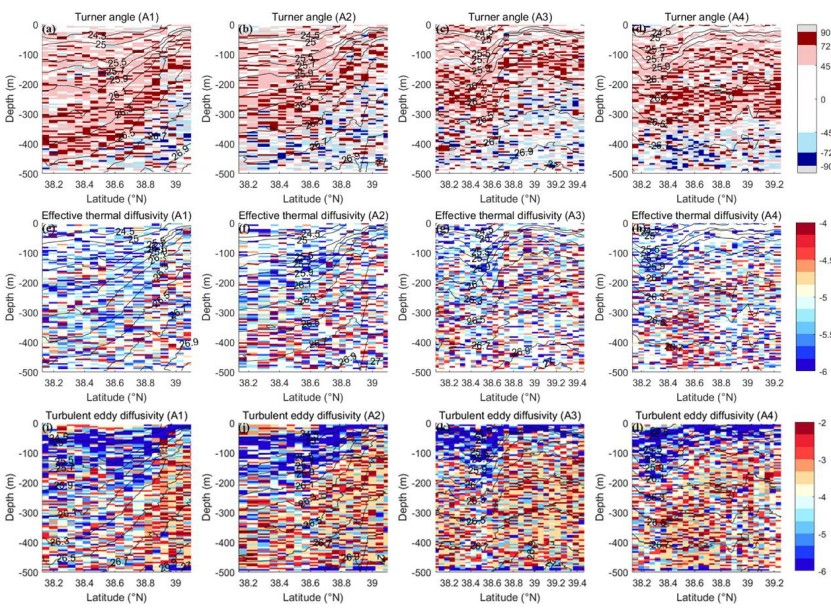

Figure 5. (a-d) Turner angle (Tu) (shading in °), (e-h) $\log_{10}$ of effective thermal diffusivity ($K_\theta$) (shading in m²/s) and (i-l) $\log_{10}$ of turbulent eddy diffusivity ($K_\rho$) (shading in m²/s) of the four sections. Contours indicate the potential density (kg/m³).

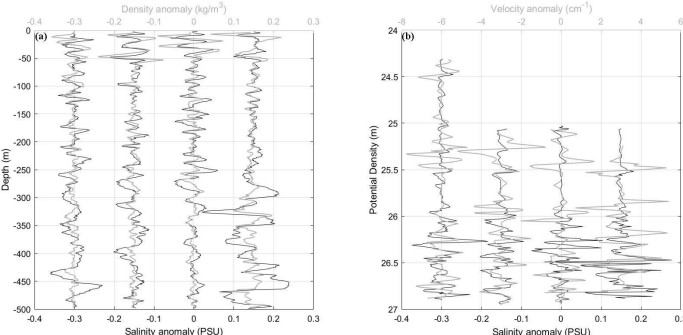

Figure 6. (a) Density anomaly and salinity anomaly, (b) velocity anomaly and salinity anomaly of the same representative profiles as figure 2. Each profile is shifted along the x axis by 0.15-PSU intervals (left to right: A1-A4).

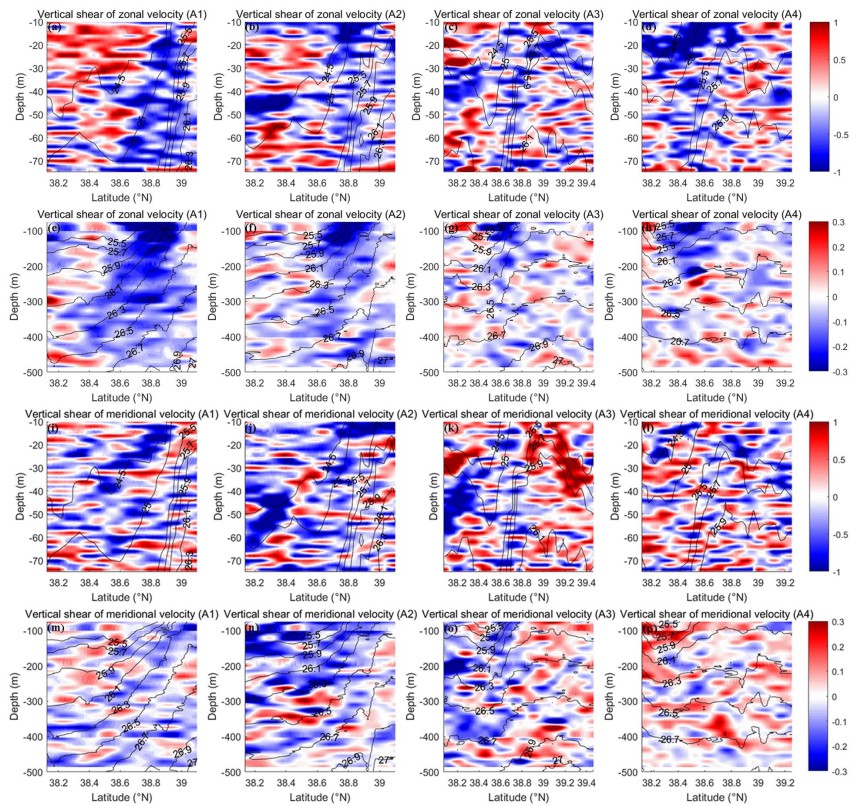

Figure 7. (a-h) Vertical shear of zonal velocity (shading in ×10⁻²/s), (i-p) vertical shear of meridional
velocity (shading in 10⁻²/s) of the four sections. Contours indicate the potential density (kg/m³).

**3.4 Double diffusion Mixing and Turbulence Mixing across the Kuroshio Extension Front**

We analyze mechanisms for the thermohaline intrusions last section. Double diffusion process and
current field instability are related to intrusions. The diapycnal mixing caused by them will be assessed
next through parameterizations, as shown in Figure 5e-l. Specific methods could be found in Section 2.
$K_\theta$ is $10^{-6}$-$10^{-4}$ m²/s. It is smaller than $10^{-5}$ m²/s in the layer upper than $\sigma_\theta$=26.3 kg/m³, and greater than
$10^{-5}$ m²/s mainly in the layer deeper than $\sigma_\theta$=26.3 kg/m³. This implies that strong diapycnal mixing
caused by double diffusion takes place in the NPIW layer where is also the primary interleaving layer.
Comparing with the distribution of Tu, we can find both of the double diffusion regime including salt
fingering and diffusive convection regime could cause strong diapycnal mixing. Our results are similar
to *Nagai et al.* [2015] that enhanced double-diffusive convection is below the main stream.
$K_\rho$ is $10^{-6}$-$10^{-2}$ m²/s. It is quite small (~$10^{-6}$ m²/s) in the layer $\sigma_\theta$=24.5-25.9 kg/m³, and big (>$10^{-4}$ m²/s) in
the layer upper than $\sigma_\theta$=24.5 kg/m³ and deeper than $\sigma_\theta$=26.3 kg/m³. The small $K_\rho$ in the mixed layer is
caused by strong mechanical stirring [*Pérez-Santosac et al.*, 2014]. Besides, turbulence is very weak
near the upper layer of fronts but strong around the upper layer of eddies' the other side interior.
Although both of the two layers have strong current shears, the former could be compensated by strong
stratification. In the interleaving layer, the $K_\rho$ is big and even beyond $K_\theta$, which may be attributed to



internal wave breaking [*Inoue et al.*, 2010; *Winkel et al.*, 2002]. It indicates turbulence mixing
dominate in intermediate layer, which is similar to *Nagai et al.* [2012] that the combination of
turbulence and subduction provide a direct pathway to form subsurface salinity minima of NPIW.

**3.5 Instability Analysis of the Kuroshio Extension Front**


Last section, we find the enhanced turbulence mixing around the upper layer of eddies' non-frontal side
interior and in intermediate layer. *D'Asaro et al.* [2011] considers the enhanced turbulence mixing is
linked with frontal instability. Hence, in this section, we analyze the frontal instability to study the
strengthening mechanism of turbulent mixing.
Symmetric instability (SI, SI extract kinetic energy from the geostrophic frontal jet and feed a turbulent
cascade to dissipation) and shear instability (Kelvin-Helmholtz instability, KI) can strengthen the
turbulent mixing [*D'Asaro et al.*, 2011; *Zhu et al.*, 2019]. Key quantity for diagnosing for SI is Ertel
Potential Vorticity (q): when q<0, a flow is unstable to SI. Key quantity for diagnosing KI is
Richardson Number (Ri): When Ri<0.25, a flow is unstable to KI. Specific calculations could be found
in Section 2.
Large/relatively large negative q exists in the upper layer of front/the NPIW layer, respectively (Figure
8a-d). Ri<0.25 is frequently observed in the upper layer of eddies' non-frontal side interior and
occasionally in NPIW layer. Therefore, the enhanced turbulence mixing in the upper layer of eddies'
non-frontal side interior is attributed to KI mainly and then SI, and in intermediate layer is attributed to
SI mainly and then KI. However, due to the strong stratification, large SI in the upper layer of frontal
zone couldn't strengthen turbulent.
We calculate the baroclinic component of Potential Vorticity ($q_{hg}$) (Figure 8e-h) and horizontal
buoyancy gradient ($|\nabla_h b|$) which is proportional to minus the density gradient (Figure 8i-l). The $q_{hg}$
arising from $|\nabla_h b|$ caused by the upward-tilted isopycnals is large negative in the frontal zone and
make a great contribution to the large negative q in the upper layer of front. However, in intermediate
layer, the barotropic component $q_v$ seems to against $q_{hg}$ to a great extent, causing relatively large q
there.



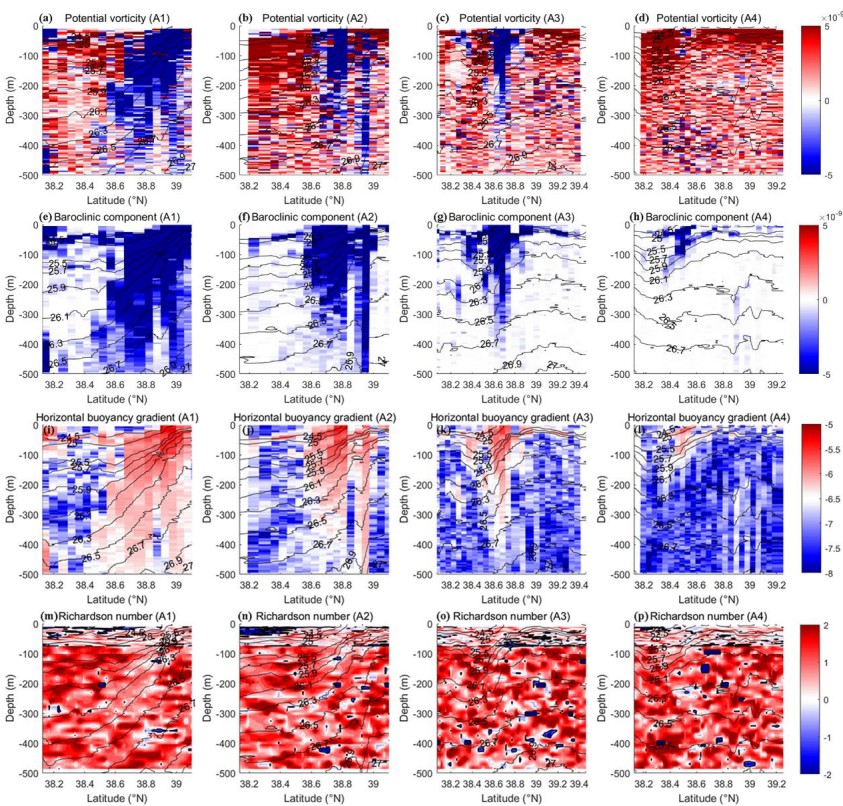

Figure 8. (a-d) Potential vorticity (shading in s$^{-3}$) and (e-h) its baroclinic component (shading in s$^{-3}$), (i-l) log$_{10}$ of horizontal buoyancy gradient (shading in s$^{-2}$) and (m-p) log$_{10}$ of Richardson number (shading) of the four sections. The region with black closed contours in (m-p) is the region with Ri <0.25. Contours indicate the potential density (kg/m3).

**4 Conclusions**

In this study, satellite remote sensing data and in situ observation data about the KEF are analyzed. The front experience a process of stable-unstable-stable state during the end of April to the end of June in 2019, which is linked with the movement of the KE's second meander. In the unstable state, the second meander transports warm and saline water to north, mix them with the cold and brackish water masses in KOCR, and cause the strong KEF. When the meander reverts to south and becomes flat, an anticyclone eddy detaches from its crest. The eddy locks and carries the KE water mass to maintain the intensity of the front. After that, it moves westward and the front becomes weak gradually.

During the period of eddy maintaining front, across front surveys including four sections are carried out. The measured thermohaline structures show the steep upward slopes of the isopycnals tilt northward in the strong frontal zone. In the layer between $\sigma_\theta$=26.5-26.7 kg/m$^3$, we observe several over 100 m thick blobs of cold and fresh water in the salinity minimum zone of NPIW. Using isopycnal anomaly method and diapycnal spiciness curvature method, characteristic interleaving layers are shown primarily in NPIW ($\sigma_\theta$=26.3-26.9 kg/m$^3$). Large variations in potential spiciness across the front seen in $\theta$-S plot and $\sigma$-$\pi$ plot illustrate that interleaving layers may arise when along-isopycnal transports occur

in intermediate layers. Furthermore, we find the thermohaline intrusions prefer to the alternate salt
fingering and diffusive convection interfaces by analysing Turner angle and are also linked with
velocity anomalies which may be caused by internal waves.
We assess the diapycnal mixing including double diffusion mixing and turbulence mixing through
parameterizations. Effective thermal diffusivity is $<10^{-5}$ m$^2$/s in the layer upper than $\sigma_\theta$=26.3 kg/m$^3$,
and $>10^{-5}$ m$^2$/s mainly in the layer deeper than $\sigma_\theta$=26.3 kg/m$^3$. Turbulent eddy diffusivity is ~$10^{-6}$ m$^2$/s
in the layer $\sigma_\theta$=24.5-25.9 kg/m$^3$, and $>10^{-4}$ m$^2$/s in the layer upper than $\sigma_\theta$=24.5 kg/m$^3$ and deeper than
$\sigma_\theta$=26.3 kg/m$^3$. Therefore, turbulence mixing dominates in intermediate layer and provide a direct
pathway to form subsurface salinity minima of NPIW. Through instability analysis, we find the strong
turbulence mixing in intermediate layer is attributed to SI (large negative q) mainly and then KI
(Ri<0.25 occasionally). The large negative q is contributed by its baroclinic component arising from
horizontal buoyancy gradient.
**Code/Data availability.** The sea surface temperature data from Optimum Interpolation Sea Surface
Temperature product are available at http://www.ncei.noaa.gov/data/sea-surface-temperature-optimum
-interpolation/access/avhrr-only/. The sea level data are available at http://marine.copernicus.eu/
services-portfolio/access-to-products/. The newly defined potential spicity functions in forms of
standard Matlab codes are available at the Supplement of *Huang et al.* [2018].
**Author contributions.** Xi Chen and Kefeng Mao collected the in situ observational data. Jiahao Wang
treated and analyzed the data. Jiahao Wang, Xi Chen and Kefeng Mao interpreted the results. Jiahao
Wang, Xi Chen, Kefeng Mao and Kelan Zhu discussed the results and wrote the paper.
**Competing interests.** The authors declare that they have no conflict of interest.
**Acknowledgments.** The authors thank NOAA and AVISO provide satellite remote sensing data for
free in their websites, and also thank all the crew members who participated in the ship cruising
observation.

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
