# Peer review of "A case study of Kuroshio Extension Front: evolution, structure, diapycnal mixing and instability"

_Ocean Science, 2020_

## Referee Comment (RC1) · Anonymous Referee #1 · 5 May 2020

Review of MS #: os-2020-10, A case study of Kuroshio Extension Front: evolution, structure, diapycnal mixing and instability. This manuscript presents results from a combined analysis of ship-based surveys and satellite observations of a front in the Kuroshio extension system. The data, which include fairly high-resolution in-situ observations from a moving-vessel profiler and multiple shipboard ADCPs, are valuable. However, the synthesis in this particular paper has some major flaws and is not a useful addition to the scientific literature in its present form. I suggest that it be rejected or withdrawn.

The main problem is that the manuscript reads like a cruise report. The analyses and conclusions are not particularly innovative, except for some results that are so remarkable as to be implausible (but the paper treats as consistent with earlier work). The

lack of novelty is not disqualifying by itself, since good observations of the processes discussed are of interest. However, since the manuscript also has several serious technical errors, which would require a long review process to fix, I recommend rejecting the paper without the option for further consideration.

The fatal flaws are as follows: 1) The diapycnal diffusivity maps K_rho are implausible (Fig. 5 bottom row). In particular, there are large areas of the main thermocline with diffusivity $\sim$10ˆ(-2) mˆ2/s. For reference, such large diffusivities are almost unobserved in the global ocean thermocline. Even the shear layer above the eastern equatorial Pacific undercurrent is almost always associated with lower diffusivities K_rho. See Peters et al. 1988 "On the parameterisation of equatorial turbulence". In addition, Nagai et al. 2015 "Evidence of enhanced double-diffusive convection..." Fig. 9d shows that K_rho only rarely gets above 10ˆ(-4) mˆ2/s and almost never gets above 10ˆ(-3) mˆ2/s.

Peters, H., Gregg, M.C. and Toole, J.M., 1988. On the parameterization of equatorial turbulence. Journal of Geophysical Research: Oceans, 93(C2), pp.1199-1218.

The paper suggests at L290 that the high K_rho could be due to internal wave breaking, but the studies cited by Inoue et al. 2010 and Winkel et al. 2002 (in the Gulf Stream and Florida current respectively) do not show such ubiquitous evidence of K_rho > 10ˆ(-3) mˆ2/s. Inoue et al. (their Fig. 4c) only observe a few isolated patches with K_rho > 10ˆ(-3) mˆ2/s, which was itself remarkable and associated with strong wintertime forcing and energetic inertial shear.

2) The potential vorticity is negative over a large region of the main thermocline below the boundary layer. This seems implausible because it would be a highly unstable state that is generally observed only in very transient and small areas near the ocean boundaries, where PV can be extracted and thereby made negative. Symmetric/inertial instability would restore the ocean interior to a state of near-zero PV in a time-scale $\sim$one day.

Thomas, L.N., 2007. Dynamical constraints on the extreme low values of the potential vorticity in the ocean. In Proc. 15th 'Aha Huliko 'a Hawaiian Winter Workshop (pp. 117-124).

3) A third issue is that statistics like correlation coefficients are not given proper uncertainty bounds.

---

## Referee Comment (RC2) · Anonymous Referee #2 · 6 May 2020

Review of the article titled "A case study of Kuroshio Extension Front: evolution, structure, diapycnal mixing and instability" by Wang et al. (MS #os-2020-10) Authors use satellite sea surface temperature and sea surface height data products as well as in situ observations of temperature, salinity and currents in the upper 500 m depth to study various aspects of Kuroshio Extension Front. The use of observational data and an attempt to conduct detailed analysis are notable features of this manuscript. However this manuscript lacks the basic requirements for a publishable research article and in its present form, I can only recommend rejecting the manuscript.

1) What is the main research question addressed in this manuscript? The lack of a central research question or focus leaves this manuscript as just "notes" prepared from an exploratory look at few datasets using few analysis methods. This is seen

throughout the manuscript, right from the abstract to the conclusions. Every figure and text describing figures just look like "show and tell" and authors do not provide any physical explanations to any of the described features. Few examples are given below.

- "Front experience a process of stable-unstable-stable state": On Line.34 authors state KEF exhibits different states on decadal time scales. Then on line 145-152, authors say the stable-unstable-stable state can be seen in few weeks of SST gradient and SLA data (Fig.1). Authors have not defined what they mean by stable and unstable state and what time scale they refer to. What causes these states? What is the relevance of these states to the current study?

- "Direct observations....indicates the steep upward slopes of the isopycnals tilt...." (from Abstract). Lines 212-213–> "The intrusions have cross-frontal orientation are laterally coherent for up to O(10) km and their vertical thickness is approximately O(100) m." Are these observed for the first time? What is the physical significance of this observation? What mechanism is active here?

2) What is the new result on features, processes, or characterization of Kuroshio Extension or Kuroshio Extension Front brought forward by this study? When presenting their analysis and what they see from their analysis, authors do not explain what was already known and what is the new aspect/feature/result their analysis brings forward. Without such a perspective, it is impossible to judge the significance of their analysis and results and the merits of their study.

3) So, in its present form, it is difficult to provide any useful comments for revising this manuscript into a publishable one. I strongly recommend the authors to follow the Jing et al. (2016) (cited in this manuscript, and has used satellite data and field observations like this study) paper as a reference, if they plan to revise this work in the future. For each section, please follow a theme as explained below from Jing et al. (2016):

Abstract: a) Clearly state what is the central focus of the study (eg. seasonal thermal fronts in the northern South China Sea). b) Major results (eg. seasonal variability of

fronts, energetic coastal upwelling/downwelling, wind forcing). c) The broad implication of results from this study (eg. importance of seasonal wind forcing in the shelf).

Introduction: a) Background (Major features of the study region like known seasonal variability, circulation, physical/dynamical processes), b) What is missing from current understanding (shelf thermal fronts and their association with seasonal winds)?, c) What is the significance of field observations here?, d) Introduce present work with its significance, major data sources, analysis methods, and explain how the rest of the paper is organized.

Data and Methods: a) Present all relevant details about a data (source, spatial and time resolution, data processing, data availability, data accuracy etc.), b) With field observations, show one dedicated figure with observations locations and major features of the region (bathymetry, currents, SSH etc), c) When explaining analysis methods, explain what a parameter measure and what mechanism/process/feature it helps to understand in the current study (see Sec. 2.3 in Jing et al. 2016)

Results: a) Explain the analysis and provide its physical interpretation, b) Clearly state the specific results from the current analysis, c) Explain whether the results agree or disagree with previous studies, d) highlight the new results from the current analysis/study, e) The transition from one analysis/section to the next one by explaining what aspect of the main scientific question is being explored in the current section.

Discussions: If there are any topics that warrant more discussion, please present it as a separate discussion section.

Summary/Conclusions: a) Summarize your main research question, data, and methods used to address the question and the major findings here. b) Provide the broad impact of the results from the current study and what aspect needs to be explored further in the future.

In summary, the current manuscript just looks like an exploratory data analysis without any specific research objective. In order to make this a publishable one, authors need to identify few research questions they can address using the available datasets, present it with proper background, formulate the analysis methods and sequence to answer specific parts of the research question and present it in relation to what is known already while highlighting the new results. I hope the comments given here will help to transform this work into a publishable one in the future.

––––––––––––––––––––––––––––